# Applying Nudge to Public Health Policy: Practical Examples and Tips for Designing Nudge Interventions

**DOI:** 10.3390/ijerph20053962

**Published:** 2023-02-23

**Authors:** Hiroshi Murayama, Yusuke Takagi, Hirokazu Tsuda, Yuri Kato

**Affiliations:** 1Research Team for Social Participation and Community Health, Tokyo Metropolitan Institute of Gerontology, Tokyo 173-0015, Japan; 2Yokohama Behavioral insights and Design Team, Kanagawa 231-0005, Japan; 3City of Yokohama, Yokohama 231-0005, Japan; 4Ministry of Finance, Tokyo 100-8940, Japan; 5PolicyGarage, Kanagawa 231-0005, Japan; 6Ministry of the Environment, Tokyo 100-8975, Japan

**Keywords:** nudge, behavioral economics, public health policy, behavioral process map, the EAST framework

## Abstract

Given the cost-effective nature of promoting desirable behaviors among individuals and societies, national and local governments have widely applied the nudge concept in various public policy fields. This viewpoint briefly explains the concept of nudge and presents the trend of nudge application in public health policy with illustrative examples. While most academic evidence on its effectiveness has been derived from Western countries, there is a significant accumulation of cases of nudge practices in non-Western countries, including the Western Pacific nations. This viewpoint also provides tips for designing nudge interventions. We introduce a simple, three-step procedure for this purpose: (1) identify target behavior, (2) determine friction and fuel of the behavior, and (3) design and implement a nudge—as well as the behavioral process map and the EAST framework.

## 1. Introduction

Many governments face severe fiscal constraints together with increasingly diverse and complex administrative challenges. In particular, the Western Pacific region is facing serious rapid ageing [1]. Therefore, the implementation of a nudge, which is a cost-effective method for promoting desirable behaviors based on behavioral economics, has become an international trend in building a healthy aging society. In fact, the utilization and application of behavioral economics to health policy in an aging society have been discussed for a long time [2], and nudge has been assimilated in geriatric care [3]. In this study, we aim to briefly explain the concept of nudge, present the trend of nudge application in public health policy and healthy aging strategies with illustrative examples, and provide tips for designing nudge interventions.

The United Nations (UN) Innovation Network [4] defines a nudge, based on Thaler and Sunstein [5], as follows: *“A behaviorally informed intervention, usually made by changing the presentation of choices (i.e., the choice architecture) to an individual, that alters people’s behavior in a predictable way. Nudges include warnings, reminders, information disclosure, simplification, and automatic enrolment. Nudges preserve freedom of choice; they do not forbid any options or significantly change economic incentives”* (p. 2).

A nudge has three main features: (1) it does not force people to engage in a particular behavior, (2) it preserves freedom of choice, and (3) it does not offer large economic incentives. Thaler and Sunstein [5] mentioned that *“to count as a mere nudge, the intervention must be easy and cheap to avoid. Nudges are not taxes, fines, subsidies, bans, or mandates. Putting the fruit at eye level counts as a nudge. Banning junk food or imposing tax on it does not”* (p. 6). This explanation helps us to understand the features of a nudge. By changing the way choices are presented, a nudge urges individuals to make more favorable decisions for themselves. A nudge is a useful tool for policy improvement because it focuses on real human behavior while considering its irrational aspects.

Nudges can be applied to various health policy domains and organizational operations including preventive healthcare, health and non-health service provision, long-term care/dementia prevention, community-based care systems, retirement planning, and technological innovation. Non-communicable diseases (NCDs) are the main targets of nudge interventions. The World Health Organization (WHO) estimated that by investing in the most cost-effective and feasible interventions to prevent and control NCDs in low- and middle-income countries, a seven-fold return could be achieved by 2030 [6]. These “best-buy” interventions could reduce tobacco and alcohol use, discourage unhealthy diets, increase physical activity, manage cardiovascular diseases and diabetes, and manage cancer. Since behavior change is the key to making most of these interventions successful, nudge interventions can contribute considerably to preventing and controlling NCDs.

In addition, effective provision of health and social services is essential for improving community health. However, it is critical to note that access to these services does not necessarily mean that people will avail themselves of them. It is well known that while health/social services are available, a considerable percentage of the population does not use them. This is referred to as the “last-mile” problem in the field of international development; although a solution has been provided, the problem is unresolved because people do not implement the solution. Nudges address this challenge precisely. By analyzing the behavioral barriers to the use of services/programs, nudges can offer interventions to overcome these barriers.

While nudges are useful policy tools, they are certainly not magic bullets. Policymakers should consider nudges as missing pieces in a set of policy tools and use them alongside traditional policy tools such as informational, financial, and regulatory approaches (Figure 1). Accordingly, it is important to understand the pros and cons of each approach, including nudging.

## 2. Effectiveness of Nudges: Evidence from a Systematic Review and Meta-Analyses

A systematic review by Ledderer et al. examined intervention studies employing nudge approaches to induce healthy behavioral changes [7]. The review included 66 studies published between 2008 and 2019 on dietary habits: five on weight control, three on physical activity, two on a combination of dietary habits and physical activity, and one on sleep patterns. These studies identified the following seven nudge techniques:(1)Accessibility of individual food items: techniques relating to accessibility, including repositioning and replacing food items;(2)Presentation of individual food items: techniques relating to food forms and amounts, food bundles, and food servings;(3)Use of messages and pictures: techniques including posters, labels, stickers and signs, pictures and thin sculptures, footprints and banners, sequences on menus, verbal prompting, and feedback;(4)Use of technology-supported information: techniques providing feedback from tracking measures, text messages, web campaigns, email, online videos, and games;(5)Use of financial incentives: techniques such as rebate programs or price reduction;(6)Use of tools affecting the senses: techniques to activate the senses such as sight, smell, and taste;(7)Cognitive loading: techniques where cognitive resources for making decisions are restricted.

Several studies have used more than one nudging technique. Moreover, Ledderer et al. noted that 42 of 66 studies reported a positive effect (i.e., nudging interventions promoted healthy behaviors), whereas three studies indicated negative effects. All negative findings involved nudges using messages and pictures, whereas one finding involved financial incentives.

While most studies included in the review were conducted in Western countries (61 of 66), a few were from non-Western areas. For example, Sim and Cheon reported the effectiveness of nudges in promoting healthy eating behaviors in Singapore [8]. They reported that salient nudge-based messages that promote healthy behaviors lead to favorable outcomes, such as improved blood sugar levels, increased vitamin intake, increased muscle mass, and reduced unhealthy dietary choices among college students. Thus, these examples suggest that nudge interventions can be applied favorably in non-Western countries as well.

The above review article further noted that dietary habits are a popular area for nudge intervention. For instance, Arno and Thomas’s meta-analysis of articles using nudge strategies to change adults’ dietary choices to healthier ones [9] indicated that nudge approaches resulted in a 15.3% increase (95% confidence interval: 7.6% to 23.0%) in healthier dietary behaviors, on average. However, Ledderer et al. reported that most of these studies were derived from Western nations (40 out of 42), mainly the United States (U.S.) [7].

Benartzi et al. conducted a rare study on the cost-effectiveness of nudges [10]. They calculated the ratio of impact-to-cost for nudge interventions and traditional policy tools such as the provision of information and financial incentives. Since there was limited availability of the required cost information, their study evaluated only four programs: retirement savings, college enrolment, energy conservation, and influenza vaccinations. In all four policy areas, nudges were more cost-effective than traditional policy tools, mainly because of the relatively low cost of interventions. For example, consider the relative effectiveness of interventions for influenza vaccinations. Here, the implementation of interventions using a planning–prompt nudge—that is, the intention to write down both the date and time of the vaccination—was effective. This nudge intervention increased the number of people vaccinated per 100 U.S. dollars spent by approximately 12.8 compared to the usual vaccination system. In other words, non-nudge interventions using traditional educational campaigns for vaccination, monetary incentives, and out-of-pocket removal increased the number of vaccinated people by approximately 8.9, 1.8, and 1.1, respectively.

There have been several findings on nudge interventions that can promote healthy behaviors. However, studies from non-Western regions, including the Western Pacific countries, remain sparse. Given that the cultural and behavioral backgrounds of these countries differ from those of Western countries, empirical findings on a nudge approach in non-Western areas are strongly warranted.

## 3. Nudge Application in Public Policy

Nudge is widely applied by national and local governments globally in various public policy fields, including health. Recently, nudge units or behavioral design teams—a team of professionals applying behavioral science to policies and social service deliveries to improve policy outcomes for citizens—have been established to support the use of nudges in many governmental and international organizations’ policies, such as the World Bank, UN, and the Organization for Economic Co-operation and Development (OECD). According to the OECD, more than 200 nudge units exist worldwide (as of August 2018) [11].

A meta-analysis by DellaVigna and Linos reported that nudge interventions implemented by nudge units are less effective than interventions that are strictly designed and implemented as part of academic research [12]. However, nudge implementation in public policy can still be considered worthwhile because it can reduce the administrative burden and implementation cost compared with conventional approaches.

In implementing nudges, it is important to consider social and cultural contexts. It is reported that not only people’s acceptance of nudges but also the effectiveness of nudges differed by country and region [13,14,15]. This implies that the generalizability of the nudge approach must be further investigated, and policymakers should carefully interpret the findings of nudges from different social/cultural contexts. Moreover, this shows the importance of understanding how nudges have been incorporated into public policy in each context (e.g., country and region) and what kind of achievements have been developed by nudge application into public policy. The following section briefly introduces the current trends in nudge applications in public policy in four leading countries: the United Kingdom (U.K.), the U.S., Singapore, and Japan.

### 3.1. The U.K.

The U.K. has led the application of behavioral science to public policy. In 2010, the U.K.’s first nudge unit, the Behavioral Insights Team (BIT), was established in the Cabinet Office. BIT has demonstrated highly cost-effective results in various fields, such as public health, energy efficiency, and tax compliance. According to BIT’s 2011–2012 annual update, it has achieved 22 times more cost savings than its running cost [16]. Owing to this tremendous success, BIT-style nudge applications have become the gold standard for nudge units worldwide.

BIT’s success has turned it into a limited company jointly owned by the U.K. government, Nesta (an innovation charity), and BIT employees. Since 2019, more than 24 government and relevant public organizations in the U.K. have established their own dedicated behavioral science teams or appointed individuals as behavioral insight specialists.

### 3.2. The U.S.

The U.S. is at the center of the world regarding behavioral science research and its applications in business, academia, and the government. Since the mid-2000s, the U.S. government has increasingly considered the application of behavioral insights, including the passing of the Pension Protection Act, which encourages opting out instead of opting into state-subsidized private pension plans. From 2009–2012, the Office of Information and Regulatory Affairs (OIRA) systematically introduced behaviorally informed policies in collaboration with other departments.

Moreover, the Obama administration incorporated behavioral science into an evidence-based policymaking initiative. In 2015, the Social and Behavioral Sciences Team (SBST) was formed in the White House to assist government agencies in applying behavioral insights and evaluating program interventions. Although the SBST charter expired in 2017, the U.S. General Services Administration (GSA) continues to support behavioral science applications in federal agencies. Multiple organizations have established teams or assigned dedicated persons to apply behavioral science concepts, including nudges, at the federal, state, and local government levels.

### 3.3. Singapore

Singapore has led the application of behavioral insights in Asia. In 2011, Singapore’s first nudge unit was established by the Ministry of the Environment and Water Resources. Since then, statutory boards of multiple ministries have established nudge units. The U.K. BIT advised Singapore’s ministries of Manpower and Transport in their behavioral science applications. Further, the National University of Singapore collaborated with the government to integrate human-centered design and behavioral science concepts into public policy. As of 2019, at least 15 government agencies in Singapore have used behavioral science techniques.

### 3.4. Japan

Japan lags behind the aforementioned three countries in the application of nudges in public policy. However, over the last few years, the momentum to apply nudges in policymaking has markedly increased among the country’s local and municipal governments. Japan’s first nudge unit was established by the Ministry of Environment in 2015, and its national Behavioral Sciences Team (BEST), which coordinates and liaisons with Japanese nudge units, was formed in 2017.

In February 2019, the Yokohama Behavioral insights and Design Team (YBiT) was founded as the first nudge unit of Japan’s local government. This unit has introduced a new dimension to policymaking at the local government level, acting as a model for other local governments. This led to the establishment of eight local government nudge units and increased the application of nudges at the municipal level across Japan.

Against this background, leaders of local nudge units established PolicyGarage in January 2021 as an incorporated non-profit organization to meet the need to apply behavioral insights and human-centered design in the public arena. PolicyGarage’s activities are not limited to Japan; for example, it has collaborated with the WHO Western Pacific Region to provide introductory hands-on training and consultation to promote healthy aging. It also collaborated with Osaka University and the Association of Behavioral Economics and Finance to provide knowledge sharing, online training programs, and consulting services to build an evidence base to incorporate behavioral insights into public policy.

## 4. Examples of Nudge Practice in Japan’s Health Policy: Focusing on Health Check-Ups

The uptake of health check-ups and instructions are critical determinants of a healthy life and healthy aging. However, various cognitive biases hinder decisions and actions regarding health checkups. While many people understand the benefits of early detection of diseases or health-related risks through regular health checkups, some may delay or avoid checkups, presenting irrationality. Behavioral science can help policymakers identify what and how cognitive biases influence people’s decision making regarding health checkups, and nudges can be an effective and cost-efficient tool to manage such biases.

Japan’s Ministry of Health, Labour, and Welfare introduced a variety of nudge interventions in a handbook for the uptake of health check-ups [17]. The town of Takahama, Fukui, presents the choice of health checkup uptake as the default rather than as an additional option. In this town, cancer screening is integrated into the general medical examination, and citizens are asked to circle the desired date of cancer screening on a form rather than being encouraged to take it. In addition, residents had to provide reasons for not undergoing screening on the form. The group that received the opt-out-type form exhibited a higher uptake rate (after the intervention: 53%) than the group that received the opt-in form (before intervention: 36%).

The city of Chiba, in the Chiba prefecture, uses a different type of nudge—simple and clear instructions—for the uptake of a general health check-up. City officials wrote a letter to local residents stressing “where to get” the check-up. Rather than ambiguously encouraging the uptake, concrete instruction to choose a hospital and make an appointment resulted in a 3.7% increase in the overall uptake rate compared with the conventional method.

The city of Hachioji, Tokyo, has improved the uptake ratio of colorectal cancer screening using a low-cost nudge intervention. The city usually sends *fecal occult blood test* kits to colorectal cancer screening recipients and mails a reminder to them after several months. A nudge technique was applied to the reminder using the prospect theory. This theory is based on loss aversion, wherein people asymmetrically feel that losses are greater than equivalent gains. Based on this theory, two types of messages were created: (A) a gain message and (B) a loss message (Figure 2). The city sent message A to 1761 people and message B to 1767 people and compared the screening uptake rates of the two groups. The uptake rate in the group that received message A was 22.7%, whereas that in the group that received message B was 29.9%. In particular, the group that received the loss-averse message had a 7.2% higher uptake rate.

## 5. Three Steps in Designing a Nudge Practice

There are several process-type frameworks for the development of nudge interventions. When policymakers implement nudges, it is better to employ clear and simple methods to help stakeholders and beneficiaries understand the concept of nudge intervention. In this section, we introduce the simplest procedure, the “three steps to design nudge”, developed based on the behavior, analysis, strategies, intervention, and change, i.e., the OECD’s BASIC framework for behaviorally informed policymaking [18]. The first step, “identify target behavior”, and the second step, “determine friction and fuel of the behavior”, utilize a behavioral process map. The third step, “design and implement nudge”, is based on the EAST framework (Figure 3).

### 5.1. Step 1: Identify Target Behavior

This step aims to appropriately define the behavior to be changed. The following three factors should be focal points:(1)Specific, such as regarding the target population, frequency, or degree of the objective behavior;(2)Meaningful enough to improve policy outcomes;(3)Measurable so that the impact of nudge intervention can be assessed.

For example, promoting NCD prevention and control is a broad target for nudge interventions. This should be broken down into specifics at the behavioral level. The target behavior is the ultimate goal of the behavioral process. Once the target behavior is identified, it can be placed at the end of a behavioral process map, which helps focus on the specific behavior that requires intervention.

### 5.2. Step 2: Determine the Friction and the Fuel of the Behavior

The second step focuses on the behavioral processes that people should undergo before achieving the target behavior. This process should be as follows:(1)As granular as possible because it is important to review whether the process can be broken down further;(2)Developed from the standpoint of the target population and not a policymaker;(3)Reviewed with relevant stakeholders to ensure that important information is not missed.

This step includes another critical task: identifying all potential barriers to and enablers of behaviors. Barriers or potential sources of friction exist that can prevent the target population from engaging in each behavior. Enablers are fuels that promote the target population’s uptake of favorable behavior. Barriers are often the inverse of enablers but not always. Figure 4 presents a behavioral process map with barriers and enablers to increase attendance at a community health center following a medical referral based on screening. This behavioral process map was developed based on the actual circumstances in the Federated States of Micronesia.

### 5.3. Step 3: Design and Implement Nudge

In the third step, a nudge intervention is designed and implemented to promote the target behavior. Nudges are designed to overcome barriers and take advantage of the enablers identified in the second step. The EAST framework can be useful in designing nudge approaches [19]. Developed by the U.K. BIT, it provides clues for designing nudges effectively by considering four key pillars for nudging: easy, attractive, social, and timely (Figure 5). Although there are several types of frameworks to design nudges, the EAST framework is relatively easy to understand and is therefore used extensively worldwide. Figure 6 illustrates an example of the EAST framework for reducing barriers and promoting enablers, based on the case shown in Figure 4.

Verifying the effectiveness of the nudge intervention is important to determine what works and how to improve it. There are several different methods of impact assessment for field experiments, such as randomized control trials and quasi-experimental studies, including propensity score matching, difference-in-differences, and regression discontinuity design. A method corresponding to a higher level of evidence is the most desirable. However, in reality, it is recommended that an appropriate method is chosen considering feasibility (e.g., local acceptance and data availability).

## 6. Ethical Considerations

The use of nudges in public policy is sometimes tricky, as care needs to be taken to ensure that the nudge is not misused to the detriment of individuals. The term “libertarian paternalism” used to describe the concept of nudges has received much criticism in recent years. This is because policymakers can use insights into public behavior to implement policies that may not be of help to most citizens. This is sometimes called “sludge”. Sludge is a tool based on cognitive biases and choice architecture, similar to a nudge. However, sludge is typically understood as friction, which makes good decisions more difficult. Moreover, they do not necessarily increase most people’s welfare—but they might benefit other entities. Therefore, the government and citizens should ensure that sludge does not creep into public policy.

In addition, the “dark side” of nudges has been sometimes discussed [20]. Some nudges, even though trying to achieve a good and helpful result for the people being nudged, can sometimes result in the opposite result. For example, suppose that a nudge with presenting social norms was given to encourage people to undergo health check-ups by showing the message “one out of every two people receives a health check-up”. Some people may feel embarrassed that they had not received a health check-up, feeling that “half the people have already attended the check-up, but I haven’t” and therefore get one. However, others may feel that “half the people are not getting health check-ups, so I don’t have to get it either” and may not be encouraged to get them. We must remember that the influence of nudges is always two-sided.

Before implementing a nudge intervention, policymakers must ensure that the nudges they intend to implement are ethically justifiable. They should be transparent about the nudge’s intent, and inform people about the process through which the nudge was developed. This should be done even though it may make the implementation less effective. In addition, they should discuss and build a consensus about the purpose of nudge intervention while identifying desirable behaviors and a desirable state of society.

Although there are different criteria and checkpoints, depending on the policy field and institution, the FORGOOD framework is recommended as the first step in examining a nudge project in light of ethical considerations (Table 1) [21].

## 7. Community Organizing

The answer to the question “What type of nudge is effective?” depends not only on the health issues of the local community but also on the community culture, history, lifestyle, and customs. Additionally, while nudges can be expected to have short-term effects, it is difficult to expect them to have long-term effects. Therefore, a system in which a variety of nudges is devised and continually implemented based on community conditions is required to promote healthy aging. Community organization is critical for this purpose.

Devising and implementing various nudges may be challenging for an individual or even a single department. To deliver necessary services to people who require them, health professionals, governments, private companies, and local residents should build coalitions and share their knowledge, experiences, expertise, and resources. They can also discuss priorities for countermeasures and ethical issues regarding nudges. This can contribute to the creation of nudges that are appropriate for local conditions.

## 8. Conclusions

Nudging is a cost-effective policy intervention approach that can potentially improve and leverage traditional policies and programs for developing a healthy aging society. It can be used to promote desirable behaviors among individuals and societies. In the future, nudges may be applied to promote healthy lifestyle habits such as diet, exercise, and sleep as well as to improve living and social environments, foster social capital, eliminate prejudice and discrimination against minorities/vulnerable populations, and enhance the accessibility of medical and nursing care resources.

In addition to applying nudges to a wide range of public health policy areas, they may be used to improve organizational efficiency. In particular, they can be used to reduce administrative burdens and improve beneficiaries’ access to various health programs. Behavioral science techniques, particularly nudges, have been widely applied in public policymaking and implementation in many countries. One reason for the increased application of nudges by governments is their ease of use. Nudge application does not necessarily require expertise; thus, individual government officials who play critical roles in the frontlines of health policy and service delivery can use it in their work.

However, nudging is not a panacea. Nudges alone do not solve complex public health problems. As Figure 1 shows, when nudges are used in conjunction with multiple approaches such as the provision of information, financial incentives, and regulations, the possibility of behavioral change increases. Thus, incorporating nudges into health policies will quickly accelerate measures that lead society in a healthier direction.

## Figures and Tables

**Figure 1 ijerph-20-03962-f001:**
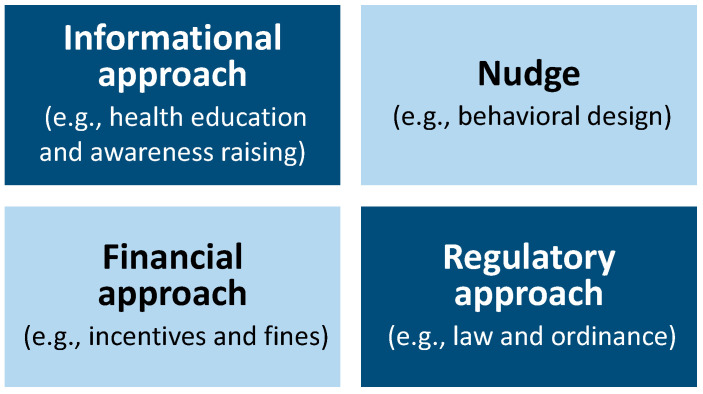
Multiple tools for policy implementation.

**Figure 2 ijerph-20-03962-f002:**
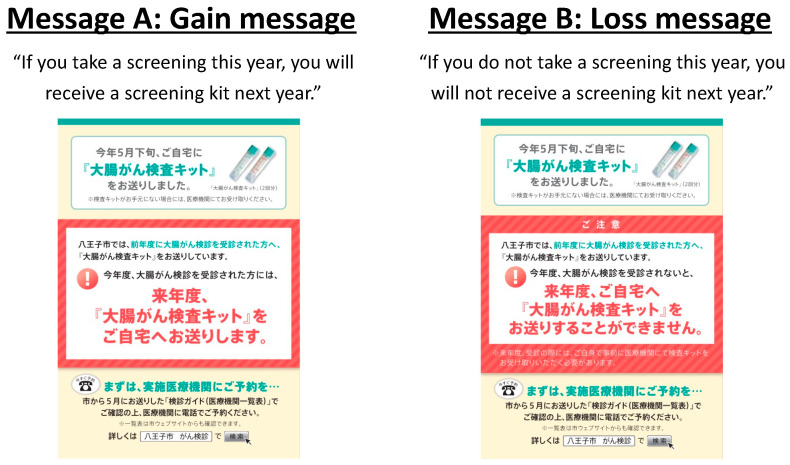
Two messages to remind citizens about the colorectal cancer screening.

**Figure 3 ijerph-20-03962-f003:**
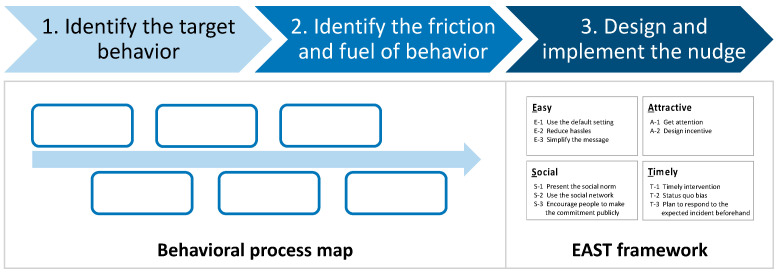
Three steps to design a nudge practice.

**Figure 4 ijerph-20-03962-f004:**
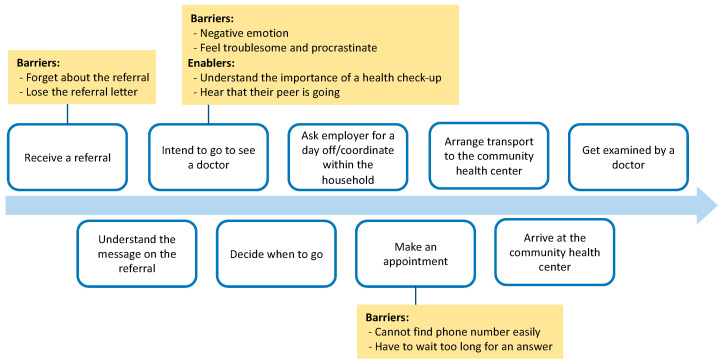
Behavioral process map and barriers/enablers: Example to increase attendance at the community health center following a medical referral based on screening. Note. There are likely more barriers and enablers for each step, but only the major elements are shown in the figure.

**Figure 5 ijerph-20-03962-f005:**
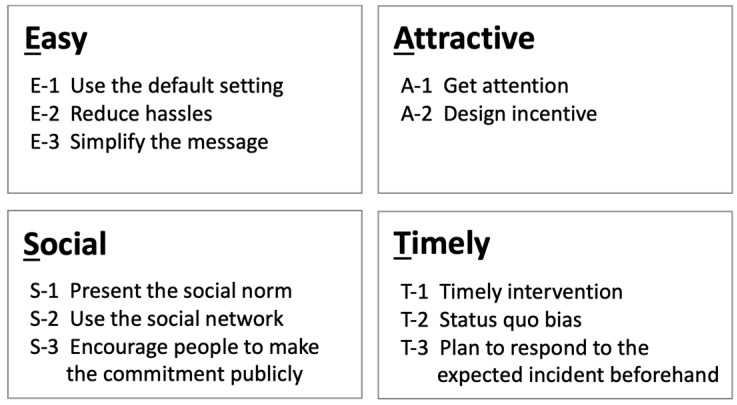
The EAST framework.

**Figure 6 ijerph-20-03962-f006:**
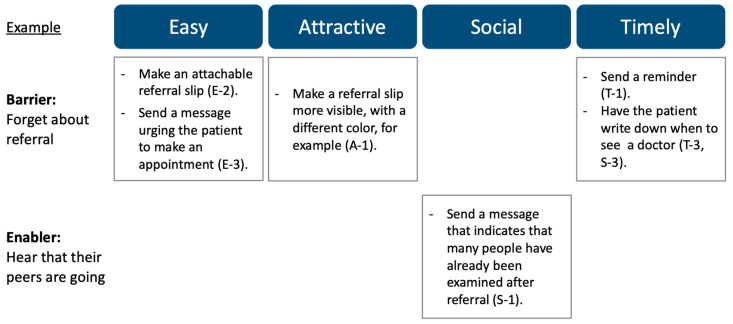
Use of the EAST framework for barriers and enablers: Example to increase attendance at the community health center following a medical referral based on screening.

**Table 1 ijerph-20-03962-t001:** The FORGOOD framework.

Key Factors	Explanation
Fairness	Does the behavioral policy have undesired redistributive effects?
Openness	Is the behavioral policy open or hidden and manipulative?
Respect	Does the policy respect people’s autonomy, dignity, freedom of choice, and privacy?
Goals	Does the behavioral policy serve good and legitimate goals?
Opinions	Do people accept the means and ends of the behavioral policy?
Options	Do better policies exist, and are they warranted?
Delegation	Do policymakers have the right and ability to nudge using the power delegated to them?

## Data Availability

Not applicable.

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
