# Peer review of "Applying Nudge to Public Health Policy: Practical Examples and Tips for Designing Nudge Interventions"

_ijerph, 2023, doi:10.3390/ijerph20053962_

Round 1
Reviewer 1 Report
1 Editorial deficiencies: lines 136-138; 167-169 contain text formatting instructions not removed from the template ("This section may be divided by subheadings. It should provide a concise and precise description of the experimental results, their interpretation, as well as the experimental conclusions that can be drawn"). These fragments should be removed.
2 What I find missing in the article is a broader reflection (and reference to existing research) on the importance of cultural and social context for nudge effectiveness. Such studies from Korea have already been published: https://journals.sagepub.com/doi/abs/10.1177/1043463120937832
Example of another publication on this topic: https://journals.sagepub.com/doi/pdf/10.1177/0011392115590086
or: https://onlinelibrary.wiley.com/doi/pdf/10.1002/cb.1861
3. Nudge has been treated very one-sidedly. The authors do not mention the negative aspects of this phenomenon (e.g., https://www.routledge.com/The-Dark-Side-of-Nudges/Madi/p/book/9780367787974)
albo: https://link.springer.com/article/10.1007/s12130-010-9129-1
I recommend supplementing the article with these threads.
Author Response
Reviewer 1
- Editorial deficiencies: lines 136–138; 167-169 contain text formatting instructions not removed from the template ("This section may be divided by subheadings. It should provide a concise and precise description of the experimental results, their interpretation, as well as the experimental conclusions that can be drawn"). These fragments should be
Response: Thank you for your kind comments. In accordance with your comment, we have removed text formatting instructions from the manuscript. We apologize for the errors.
- What I find missing in the article is a broader reflection (and reference to existing research) on the importance of cultural and social context for nudge effectiveness. Such studies from Korea have already been published: https://journals.sagepub.com/doi/abs/10.1177/1043463120937832.
Example of another publication on this topic: https://journals.sagepub.com/doi/pdf/10.1177/0011392115590086
https://onlinelibrary.wiley.com/doi/pdf/10.1002/cb.1861
Response: We agree with this suggestion. A description of the effectiveness of nudges according to social and cultural contexts was missing in the previous version of the manuscript. We have explained this in the revised manuscript (lines 152–159).
“In implementing nudges, it would be important to consider social and cultural contexts. It is reported that not only people’s acceptance of nudges but also the effectiveness of nudges differed by country and region [13–15]. This implies that the generalizability of the nudge approach must be further investigated, and policymakers should carefully interpret the findings of nudges from different social/cultural contexts. Also, this shows the importance of understanding how nudges have been incorporated into public policy in each context (e.g., country and region) and what kind of achievements have been developed by nudge application into public policy.” (lines 152–159)
- Nudge has been treated very one-sidedly. The authors do not mention the negative aspects of this phenomenon (e.g., https://www.routledge.com/The-Dark-Side-of-Nudges/Madi/p/book/9780367787974)
albo: https://link.springer.com/article/10.1007/s12130-010-9129-1
I recommend supplementing the article with these threads.
Response: In accordance with your advice, we have added descriptions regarding the dark side of nudges (lines 337–346).
“In addition, the “dark side” of nudges has been sometimes discussed [20]. Some nudges, even though trying to achieve a good and helpful result for the people being nudged, can sometimes result in the opposite result. For example, suppose that a nudge with presenting social norms was given to encourage people to undergo health check-ups by showing the message, "one out of every two people receives a health check-up." Some people may feel embarrassed that they had not received a health check-up, feeling that "half the people have already attended the check-up, but I haven't," and therefore get one. However, others may feel that "half the people are not getting health check-ups, so I don't have to get it either," and may not be encouraged to get them. We must remember that the influence of nudges is always two-sided.” (lines 337–346)

Reviewer 2 Report
Excellent contribution.
Just a suggestion. The following paragraph is repeated twice on page 4.
"This section may be divided by subheadings. It should provide a concise and precise description of the experimental results, their interpretation, as well as the experimental conclusions that can be drawn."
Author Response
Reviewer 2
- Just a suggestion. The following paragraph is repeated twice on page 4. "This section may be divided by subheadings. It should provide a concise and precise description of the experimental results, their interpretation, as well as the experimental conclusions that can be drawn."
Response: Thank you for your kind comments. We have removed these sentences from the article. In the revised manuscript, we carefully checked and removed them.

Reviewer 3 Report
I find the article very interesting and of great scientific contribution to apply to ageing societies in order to improve the global society. Congratulations, the paper is simple, clear and straightforward. However, I miss more bibliographical references on the subject.
If you would search for more information in research search engines, the manuscript would improve a lot.
Thank you.
Author Response
Reviewer 3
- I find the article very interesting and of great scientific contribution to apply to ageing societies in order to improve the global society. Congratulations, the paper is simple, clear, and straightforward. However, I miss more bibliographical references on the subject.If you would search for more information in research search engines, the manuscript would improve a lot.
Response: Thank you for your kind comments. In accordance with your comment, we have added some references regarding nudges and aging society/geriatric care to the revised manuscript (lines 30–32).
“In fact, the utilization and application of behavioral economics to health policy in an aging society have been discussed for a long time [2], and nudge has been assimilated in geriatric care [3].” (lines 30–32)

Round 2
Reviewer 1 Report
The authors have made the necessary corrections. I am pleased to say that the article can be published.